



# Improved Advection, Resolution, Performance, and Community Access in the New Generation (Version 13) of the High Performance GEOS-Chem Global Atmospheric Chemistry Model (GCHP)

Randall V. Martin[1], Sebastian D. Eastham[2], Liam Bindle[1], Elizabeth W. Lundgren[3], Thomas L. Clune[4],
Christoph A. Keller[4,5,†], William Downs[3,&], Dandan Zhang[1], Robert A. Lucchesi[4], Melissa P. Sulprizio[3],
Robert M. Yantosca[3], Yanshun Li[1], Lucas Estrada[3], William M. Putman[4], Benjamin M. Auer[4], Atanas
L. Trayanov[4], Steven Pawson[5], Daniel J. Jacob[3]

[1]Department of Energy, Environmental & Chemical Engineering, Washington University in St. Louis, St. Louis, MO, USA

[2]Laboratory for Aviation and the Environment, Massachusetts Institute of Technology, Cambridge, MA, USA

[3]John A. Paulson School of Engineering and Applied Sciences, Harvard University, Cambridge, MA, USA

[4]Global Modeling and Assimilation Office, NASA Goddard Space Flight Center, Greenbelt, MD, USA

[5]Universities Space Research Association, Columbia, MD, USA.

[†]Now at Morgan State University, MD, USA

[&]Now at Rosenstiel School of Marine and Atmospheric Science, University of Miami, FL, USA

*Correspondence to*: Randall V. Martin (rvmartin@wustl.edu)

**Abstract.** We describe a new generation of the high-performance GEOS-Chem (GCHP) global model of atmospheric composition developed as part of the GEOS-Chem version 13 series. GEOS-Chem is an open-source grid-independent model that can be used online within a meteorological simulation or off-line using archived meteorological data. GCHP is an
offline implementation of GEOS-Chem driven by NASA Goddard Earth Observing System (GEOS) meteorological data for massively parallel simulations. Version 13 offers transformational advances in GCHP for ease of use, computational performance, versatility, resolution, and accuracy. Specific improvements include (a) stretched-grid capability for higher resolution in user-selected regions, (b) easier build with a build system generator (CMake) and a package manager (Spack), (c) software containers to enable immediate model download and configuration on local computing clusters, (d) better
parallelization to enable simulation on thousands of cores, (e) multi-node cloud capability, and (f) more accurate transport with new native cubed-sphere GEOS meteorological archives including air mass fluxes at hourly temporal resolution with spatial resolution up to C720 (~12 km). The C720 data are now part of the operational GEOS Forward Processing (GEOS-FP) output stream, and a C180 (~50 km) consistent archive for 1998-present is now being generated as part of a new GEOS-IT data stream. Both of these data streams are continuously being archived by the GEOS-Chem Support Team for access by
GCHP users. Directly using horizontal air mass fluxes rather than inferring from wind data significantly reduces global mean error in calculated surface pressure and vertical advection.



## 1 Introduction

Atmospheric chemistry and composition are central drivers of climate change, air quality, and biogeochemical cycling. They are a next frontier for Earth system model (ESM) development (NRC, 2012). Modeling of atmospheric chemistry is a
grand scientific and computational challenge because of the need to simulate hundreds of gaseous and aerosol chemical species stiffly coupled to each other and interacting with transport on all scales. There is considerable demand for high-resolution atmospheric chemistry models from a broad community of researchers and stakeholders with interest in simulating a range of problems at local to global scales. But software engineering complexity and computational cost have been major barriers to access.

Atmospheric chemistry models solve the 3-D continuity equations for an ensemble of reactive and coupled gaseous/aerosol chemical species with terms to describe emissions, transport, chemistry, aerosol microphysics, and deposition (Brasseur and Jacob, 2017). The model may be integrated "online" within a meteorological model or ESM, with the chemical continuity equations solved together with the equations of atmospheric dynamics, or "offline" as a chemical transport model (CTM) where the chemical continuity equations are solved using external meteorological data as input. The online approach has the
advantage of more accurately coupling chemical transport to dynamics, and has specific application to the study of aerosol-chemistry-climate interactions. It also enables consistent chemical and meteorological data assimilation. The offline approach has advantages of accessibility, cost, portability, reproducibility, and straightforward application to inverse modeling. The broad atmospheric chemistry community can easily access an offline CTM for reusable applications that advance atmospheric chemistry knowledge, but access to an online model is more limited and complicated. Ideally, the same
state-of-the-art model must be able to operate both online and offline.

The GEOS-Chem atmospheric chemistry model (geos-chem.org) delivers this joint online-offline capability. GEOS-Chem is an open-source global 3-D model of atmospheric composition used by hundreds of research groups around the world for a wide range of applications. It simulates tropospheric and stratospheric oxidant-aerosol chemistry, aerosol microphysics, carbon gases, mercury, and other species (e.g., Eastham et al., 2014; Friedman et al., 2014; Kodros and Pierce, 2017; Li et
al., 2017; Shah et al., 2021; Wang et al., 2021). GEOS-Chem has been developed and managed continuously for the past 20 years (starting with Bey et al. (2001)) as a grass-roots community effort. The online version is part of the Goddard Earth Observation System (GEOS) of the NASA Global Modeling and Assimilation Office (GMAO) (Long et al., 2015; Hu et al., 2018; Keller et al., 2021) and has been implemented in other climate and meteorological models as well (Lin et al., 2020; Lu et al., 2020; Feng et al., 2021). The offline version uses exactly the same scientific code and is driven by GEOS
meteorological data or by other meteorological fields (Murray et al., 2021). The offline GEOS-Chem has wide appeal among atmospheric chemists because it is a comprehensive, cutting-edge, open-source, well-documented modeling resource that is easy to use and modify but also has strong central management, version control, and user support through a GEOS-Chem Support Team (GCST) based at Harvard University and at Washington University.





The standard offline version of GEOS-Chem ("GEOS-Chem Classic") is designed for easy use and a simple code base but
relies on shared-memory parallelization and a rectilinear longitude-latitude grid, limiting its flexibility and scalability for
high-resolution applications in modern High Performance Computing (HPC) environments. A high-performance version of
GEOS-Chem (GCHP) was developed by Eastham et al. (2018) to address this limitation. GCHP is a grid-independent
implementation of GEOS-Chem using Message Passing Interface (MPI) distributed-memory parallelization enabled through
the Earth System Modeling Framework (ESMF, earthsystemmodeling.org) and the Modeling Analysis and Prediction Layer
(MAPL), in the same way as the GEOS system (Suarez et al., 2007; Long et al., 2015; Eastham et al., 2018; Hu et al., 2018).
GCHP operates on atmospheric columns as its basic computation units, with grid information specified at runtime through
ESMF. Chemical transport is simulated using a finite volume advection code (FV3), allowing GEOS-Chem simulations to be
performed on the native GEOS cubed-sphere grid (Putman and Lin, 2007), but the scientific code base is otherwise the same
as GEOS-Chem Classic. GCHP enables GEOS-Chem simulations to be conducted with high computational scalability on up
to a thousand cores (Eastham et al., 2018; Zhuang et al., 2020), so that global simulations of stratosphere–troposphere
oxidant–aerosol chemistry with very high resolution become feasible.

Here we describe development of a new generation of GCHP (version 13) for improved community access, performance,
resolution, and accuracy. Section 2 provides background on GEOS-Chem and GCHP. The MAPL coupler and GEOS system
are described in Section 3. Section 4 provides a high-level overview of developments in the GCHP version 13 series. A
stretched grid capability is described in Section 5. Advances in ESM coupling and software collaboration are presented in
Section 6. In Section 7, improvements to GCHP performance and portability are described including (a) easier build system,
(b) software containers to facilitate download and configuration, (c) better parallelization, and (d) capability for multi-node
simulations on the cloud. Section 8 presents extension of GCHP capabilities to utilize fine-scale cubed-sphere meteorology
at the global scale, and describes two new cubed-sphere data streams. A performance demonstration in Section 9 is followed
by a section on future needs and opportunities. This new generation of GCHP is extensively documented on our GCHP Read
The Docs site (https://gchp.readthedocs.io/en/latest/).

## 2 GEOS-Chem and GCHP

GEOS-Chem simulates the evolution of atmospheric composition by solving the system of coupled continuity equations for
an ensemble of $m$ species (gases or aerosols) with concentration vector $\mathbf{n} = (n_1, ..., n_m)^T$:

$$\frac{\partial n_i}{\partial t} = -\nabla \bullet (n_i \mathbf{U}) + P_i(\mathbf{n}) - L_i(\mathbf{n}) + E_i - D_i \qquad i \in [1, m] \quad (1)$$

Here $\mathbf{U}$ is the wind vector (including subgrid components parameterized as boundary layer mixing and wet convection);
$P_i(\mathbf{n})$ and $L_i(\mathbf{n})$ are the local production and loss rates of species $i$ from chemistry and/or aerosol microphysics, which depend
on the concentrations of other species; and $E_i$ and $D_i$ represent emissions and deposition. Equation (1) is solved by operator
splitting of the transport and local components over finite time steps. The local operator,



$$\frac{dn_i}{dt} = P_i(\mathbf{n}) - L_i(\mathbf{n}) + E_i - D_i \qquad\qquad i \in [1, m] \quad (2)$$


includes no transport terms and thus reduces to a system of coupled first-order ordinary differential equations (ODEs). We refer to it as the GEOS-Chem chemical module even though it also includes terms for emission, deposition, and aerosol microphysics.

GEOS-Chem includes routines to conduct all of the operations in equation (1). The simulations can be conducted either
offline or online. The offline mode uses archived meteorological data, including $\mathbf{U}$ and other variables, to solve equation (1). This includes transport modules for grid-resolved advection, boundary layer mixing, and wet convection. The online mode uses the GEOS-Chem chemical module (equation (2)) to solve for the local evolution of chemical species within a meteorological model where transport of the chemical species is done independently as part of the model dynamics. The GEOS-Chem transport modules are then disabled.

The standard offline implementation of GEOS-Chem uses NASA GEOS meteorological archives as input, currently either from the Modern-Era Retrospective analysis for Research and Applications, Version 2 (MERRA-2) for 1980 to present, or from the GEOS Forward Processing (GEOS-FP) product generated in near-real-time. In GEOS-Chem Classic, first described by Bey et al. (2001), the model provides a choice of rectilinear latitude-longitude Eulerian grids with shared-memory parallelization. The coding architecture is simple but efficient parallelization is limited to a single node with tens of cores.
GEOS-Chem Classic can be used in principle at the native resolutions of MERRA-2 (0.5º×0.625º) or GEOS-FP (0.25º×0.3125º), but global simulations are limited in practice to 2º×2.5º or 4º×5º horizontal resolution because of the inefficient parallelization and prohibitive single-node memory requirements. Native-resolution simulations can be conducted for regional/continental domains (Zhang et al., 2015; Li et al., 2021), with boundary conditions from an independently conducted coarse-resolution global simulation. However, with simulated atmospheric chemistry continuously increasing in
computational complexity and the performance of individual computatational nodes relatively stagnant, the restrictions of running on a single node increasingly force users to choose between speed, resolution, and accuracy.

GCHP, first described by Eastham et al. (2018) evolved the offline implementation of GEOS-Chem to a grid-independent formulation with MPI distributed-memory parallelization. The grid-independent formulation of GEOS-Chem, originally developed by Long et al. (2015) for online applications, enables the model to operate on any horizontal grid specified at run
time. The model solves for the chemical module (equation (2)) on 1-D vertical columns of the user-specified grid, and passes the updated concentrations at each time step to the transport modules. In GCHP, this grid-independent formulation is exploited in an offline mode with the MAPL coupler and ESMF to operate GEOS-Chem on the native cubed-sphere of the GEOS meteorological model. MAPL/ESMF delivers the MPI capability allowing for efficient parallelization on up to a thousand cores (Eastham et al., 2018; Zhuang et al., 2020) and enables global simulations at the native resolution of the
GEOS meteorological data. At the same time, GCHP can still be run on a single node with similar performance as GEOS-Chem Classic for low-resolution applications.





Figure 1 shows the general architecture of GCHP. MAPL couples the different components of the model (gridded components), providing and receiving inputs, and handles parallelization. Meteorological and other data are read through the ExtData module and re-gridded as needed to the desired cubed-sphere resolution. Advection on the cubed-sphere is done with the offline FV3 module of Putman and Lin (2007). GEOS-Chem updates the chemical concentrations over model time steps in 1-D columns corresponding to the model grid, including subgrid vertical transport from boundary layer mixing and wet convection. Model output diagnostics are archived through the History module. GCHP is written in Fortran with the option to use either Intel or GNU compilers. Beyond the NetCDF libraries required for GEOS-Chem, GCHP's additional dependencies (external standalone libraries) are an MPI implementation and ESMF.

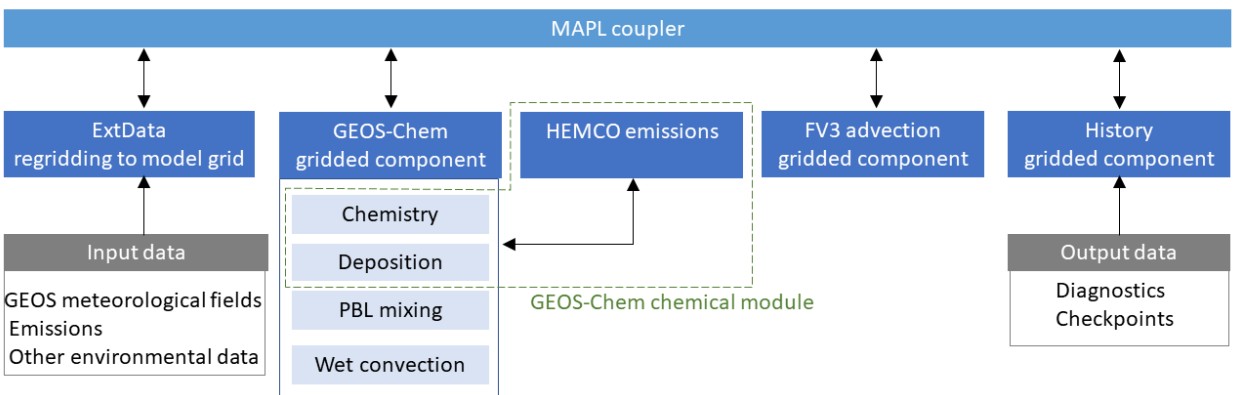

**Figure 1: Schematic of GCHP architecture. The model consists of four gridded components (ExtData, GEOS-Chem, FV3, History) exchanging information through the MAPL coupler. The HEMCO emissions module communicates directly with the GEOS-Chem gridded component in the current GCHP architecture, but it can also be used as a separate gridded component in other model architectures (Lin et al., 2021). The GEOS-Chem gridded component includes planetary boundary layer (PBL) mixing and wet convective transport of species as governed by the GEOS meteorological fields passed through MAPL. 'Chemistry' also includes aerosol microphysical processes for which the continuity equations are analogous. The GEOS-Chem chemical module as defined in the text and illustrated in the Figure includes emissions, chemistry, and deposition and would be the unit passed to a meteorological model or ESM in online applications.**

## 3 MAPL and GEOS

### 3.1 MAPL overview

MAPL is an infrastructure layer that leverages ESMF to provide services that simplify the process of coupling model components and enforce certain consistency conventions across components. In particular, MAPL provides high-level interfaces that allow developers of gridded components to readily specify the imports, exports, and internal states for their components as well as to hierarchically incorporate "child" components. The "generic" layer in MAPL translates the high-level specifications to register initialize/run/finalize methods with ESMF, allocate storage, create ESMF fields and states,



and enable the use of shared pointers wherever possible to reduce memory and performance overheads. This generic layer additionally provides common services across components such as checkpoint and restart.

MAPL also provides two highly-configurable ESMF components: ExtData and History which manage spatially distributed
input and output respectively as described in Section 2. MAPL automatically aggregates all component exports to make them available to the History component for output and sends any unsatisfied imports to ExtData for input. ExtData and History have the capability to automatically regrid to and from the model and component grid with a variety of temporal sampling and horizontal interpolation options.

MAPL also fills some gaps in ESMF functionality, though the nature of those gaps continually evolves as both frameworks
advance.  Currently MAPL provides at least one regridding method not yet available in ESMF, namely the ability to regrid horizontal fluxes in an exact manner for integral grid resolution ratios.

A major performance bottleneck in the original version of GCHP as described in Eastham et al. (2018) was in the reading of input data. The current version of MAPL includes optimizations to the ExtData layer used for input with elimination of redundant actions and use of multiple cores on a single node for data input, thereby reducing the input computational cost.
GCHP timing tests with this new capability are presented in section 7.5.

### 3.2. GEOS system

The GEOS system of NASA GMAO provides meteorological inputs needed by GEOS-Chem including wind and pressure information, humidity and precipitation data, as well as surface quantities such as soil moisture, friction velocity and skin temperature.    The    full    list    of    meteorological    input    data    used    by    GEOS-Chem    can    be    found    at
http://wiki.seas.harvard.edu/geos-chem/index.php/List_of_GEOS-FP_met_fields.

The GEOS-FP and MERRA-2 data products used to drive GEOS-Chem are generated by the GEOS ESM and Data Assimilation System (DAS), consisting of a suite of modular model components connected through the ESMF/MAPL software interface (Todling and Akkraoui, 2018). GEOS-FP (Lucchesi, 2017) uses the most recent validated version of the GEOS ESM system and produces meteorological and aerosol analyses and forecasts in near real time. Currently (version
5.27.1), it runs on a cubed-sphere grid with a horizontal resolution of C720 (approx. 12×12 km$^2$) where the resolution of the cubed-sphere output is indicated by CN, and N is the number of grid boxes on one edge of one face of the cubed-sphere. Thus the total number of cells in one model level is 6N$^2$. The outputs from this system have been conventionally archived on a latitude-longitude grid with a horizontal resolution of 0.25º × 0.3125º, incurring loss of resolution and accuracy in vector fields as presented in Section 8. MERRA-2 is a meteorological and aerosol reanalysis from 1979 to present produced with a
stable version of GEOS (Gelaro et al., 2017). MERRA-2 simulations are conducted at a lower horizontal resolution than GEOS-FP (C180 vs. C720), and all MERRA-2 fields are archived on a latitude-longitude grid with a horizontal resolution of 0.5º × 0.625º. For both GEOS-FP and MERRA-2, traditional archival has been at one-hour temporal resolution for surface quantities and three-hour temporal resolution for 3D quantities such as winds. Winds are defined at the center of the grid cell (A-Grid staggering using the notation introduced by Arakawa and Lamb (1977)), while mass fluxes are defined at the center





of the relevant grid edges in 2D contexts and interfaces of the discrete volumes of the grid in 3D contexts (C-Grid staggering) as further described in Section 8.2. New cubed-sphere archives including mass fluxes are described in Section 8.3. The GEOS-Chem Support Team historically reprocessed GEOS data into specific input formats including coarser resolution and nested domains for use by GEOS-Chem Classic. The FlexGrid option implemented in GEOS-Chem version 12.4 enabled the generation of coarse-grid and custom nested data on the fly at run time (Shen et al., 2021b) but other

reprocessing of the native fields was still required for GCHP. Access to pre-generated coarse-grid archives (2°×2.5° and 4°×5°) and pre-cut nested domains is still supported for GEOS-Chem Classic, but the reprocessing can now be skipped for GCHP as described in Section 8.4.

## 4 Overview of new capabilities in the GCHP version 13 series

Table 1 contains an overview of the new capabilities for GCHP that have been implemented as part of the version 13 series
for improved community access, performance, resolution, and accuracy.

**Table 1: Overview of new capabilities in the GCHP version 13 series**

| Feature | Section[a] |
| --- | --- |
| Stretched-grid capability for higher resolution in user-selected regions | 5 |
| Advances in ESM coupling and software collaboration | 6 |
| Easier build with a build system generator (CMake) and a package manager (Spack) | 7.1 & 7.2 |
| Software containers to enable immediate model download and configuration on local computing clusters | 7.3 |
| Improved error and output diagnostics | 7.4 |
| Better parallelization to enable simulations on thousands of cores | 7.5 |
| Multi-node cloud capability | 7.6 |
| More accurate transport through use of mass fluxes on the cubed-sphere grid | 8.1 & 8.2 |
| New hourly native cubed-sphere GEOS meteorological archives | 8.3 |
| Direct ingestion of GEOS meteorological archives | 8.4 |

[a] Section of the manuscript where the new capability is discussed

Community access was facilitated by improving the build system through a build system generator (CMake) and a package manager (Spack), by offering software containers, by improving error and output diagnostics, and by developing a multi-node cloud capability. Use of the CMake build system generator described in section 7.1 (a) improved the robustness of the build, (b) improved the maintainability of the build system, and (c) made building GCHP easier for users. Use of the Spack



package manager described in section 7.2 eased the installation of GCHP by specifying precisely how to build GCHP for
different versions, configurations, platforms, and compilers. Use of containers enabled immediate download and
configuration for cloud environments and for local environments that support containers, as described in section 7.3.
Improved error and output diagnostics facilitate debugging and evaluation, as described in section 7.4. The ability for GCHP
to directly use GEOS meteorological archives opened up new opportunities for near-real-time simulations as presented in
section 8.4.

Performance was improved through better parallelization as described in Section 7.5, enabling efficient simulations on
thousands of cores. For example, a full-chemistry simulation at C360 (~25 km) now achieves 20 model days per day of
actual time on 2304 cores. The improved parallelization was achieved by updating the MAPL software to take advantage of
improvements in input efficiency that eliminated the previous computational bottleneck as described in Section 3.1.

Resolution was improved through the generation of hourly GEOS archives for advection variables, with resolution up to
cubed-sphere C720 (~12 km) and development of a stretched-grid capability for regional refinement. The GEOS-FP C720
advection archive began production on March 11, 2021 and is continuing operationally. Hourly archiving (instead of three-
hourly previously) of the advection variables (air mass fluxes, specific humidity, Courant numbers, and surface pressure)
significantly reduces transport errors associated with transient (eddy and convective) advection (Yu et al., 2018). The cubed-
sphere archive is most critical for advection variables since they increase the accuracy of the transport simulation. An in-
progress GEOS-IT archive for the period 1998-present includes cubed-sphere archives of all meteorological variables at
hourly C180 resolution as described in Section 8.3. The stretched-grid capability was described in Bindle et al. (2021) and is
summarized in Section 5.

Accuracy was improved by directly ingesting mass fluxes instead of winds as described in Section 8.1, by conducting
simulations directly on the native cubed-sphere grid of the meteorology, and through high- resolution meteorological
archives. The use of mass fluxes is particularly important for accurate vertical transport in the stratosphere where weak
vertical motion increases susceptibility to errors from use of winds. Conducting simulations directly on the cubed-sphere
reduces errors from regridding to and from a latitude-longitude grid to and from cubed-sphere grid, and from restaggering to
and from the center of a grid cell to and from the center of a grid edge as described in section 8.2.

## 5 Stretched grid

A limitation of the original version of GCHP was the absence of a grid-refinement capability over regions of specific
interest. GEOS-Chem Classic has a nested-grid capability to allow native-resolution simulations over regional or continental
domains with dynamic boundary conditions from the global simulation (Wang et al., 2003), and these domains can be
defined at runtime with the FlexGrid facility (Li et al., 2021). This is not possible in GCHP because there is not yet a
mechanism to specify boundary conditions in a non-global domain and because FlexGrid is only for latitude-longitude grids.
Bindle et al. (2021) implemented grid stretching as a means for regional grid refinement in GCHP. Grid-stretching in GCHP



uses a modified Schmidt (1977) transform (Harris et al., 2016) to "stretch" the cubed-sphere grid for all input data through ExtData to create a refinement. The user has nimble control over the refinement location and strength using three runtime parameters; the *stretch-factor* controls the refinement strength, and the *target-latitude* and *target-longitude* control the refinement location.

Several recent developments enabled the implementation of the stretched-grid capability in GCHP. Harris et al. (2016) developed the stretched-grid capability for the FV3 advection code used in GCHP. The MAPL framework added support for that capability. An archive of state-dependent emissions at native resolution was developed for GEOS-Chem, producing consistent emissions regardless of the model grid (Weng et al., 2020; Meng et al., 2021).

Figure 2 shows a visualization of the stretched grid. A key advantage of grid-stretching compared to other refinement
techniques, such as nesting, is the smoothness of the transition from the region of interest to the global background. Stretching does not change the logical structure (topology) of the grid, and two-way coupling is inherent; this means stretching can be implemented without major structural changes to the model or the need for a component to couple the simulation across distinct model grids.

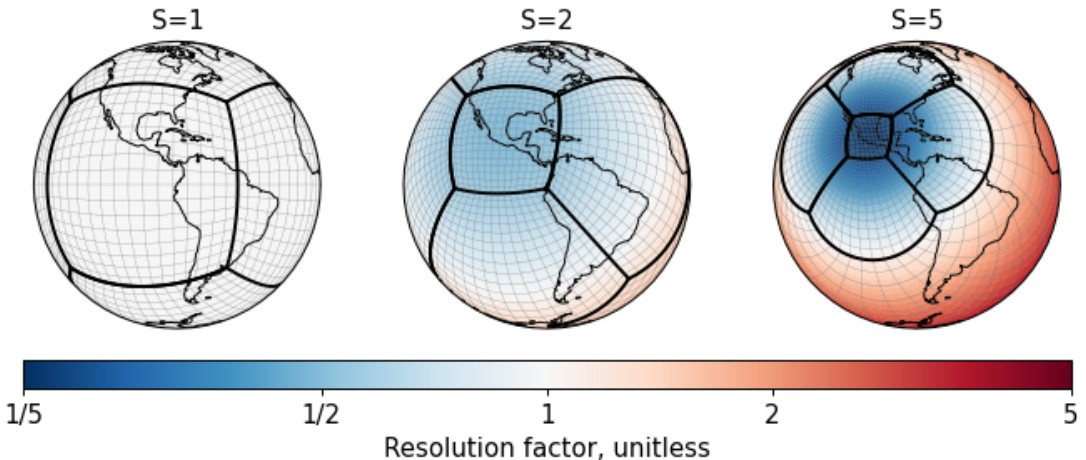


**Figure 2: Visualization of grid stretching for two refinement scenarios using stretch factors (S) on a C16 cubed-sphere. Resolution factor is the relative change of grid box edge length induced by stretching. Adapted from Bindle et al. (2021).**

## 6 Advances in ESM coupling and software collaboration

Here we describe restructuring of GCHP and its interfaces with MAPL and its dependencies in the version 13 series. These
developments were motivated by the needs for tighter coupling of GCHP with the parent ESM (GEOS), for coordinated development of MAPL between the GCHP and GMAO development teams, and for reduction in GCHP build time.

The original version of GCHP (Eastham et al., 2018) was implemented as a code repository that users manually inserted into the GEOS-Chem code base. It contained a copy of MAPL and other GMAO software libraries that were frozen during the



initial development process and contained no GMAO development history. Ongoing improvements to the MAPL
infrastructure by GMAO were not regularly or easily propagated to the GCHP code, while bug fixes and MAPL
enhancements made by GCHP developers were not easily propagated back to GMAO. This disconnect resulted in
divergence of code, difficulty updating GEOS-Chem in GEOS, and limitations on the progress of GCHP capabilities.

We restructured GCHP in version 13.0.0 to address these issues. We implemented independently maintained code bases such
as MAPL and HEMCO as Git submodules that contained version history information. We replaced in GCHP the existing
version of MAPL and its dependencies with the latest stable version releases, thereby expanding infrastructure capabilities
for GCHP such as updates necessary for simulations on a stretched grid. We also developed a system for seamless version
updates between GCHP and GEOS code bases by using forks of GMAO software repositories as Git submodules for
straightforward merging of code updates via GitHub pull requests while retaining all version history.

GCHP 13.0.0 also changed how GCHP interfaces with the ESMF library. GCHP originally contained a copy of ESMF
without version history and users were required to build ESMF from scratch with every new GCHP download. This caused
build times substantially longer than necessary given ESMF in GCHP rarely changed. To reduce build time, we restructured
GCHP to use ESMF as an external library. Users now may download ESMF from its public repository
(earthsystemmodeling.org), build it locally, and use the same build for GCHP or any other ESMF-based applications. ESMF
can even be built as a system-wide module, enabling all users on a system to use a centrally-maintained copy in the same
way that components such as compilers, NetCDF, or MPI are treated. This is beneficial for the following reasons: (a) Allows
greater flexibility for ESMF version updates, (b) Cuts initial GCHP build time in half, (c) Provides greater transparency in
ESMF via original Git history, (d) Reflects that ESMF is a separate project from GCHP, with its own model development
activities and support team, and (e) Leverages the advantages of centrally-maintained libraries in modern HPC systems,
allowing science-focused users to build and run GCHP with minimal effort.

Overall the GCHP 13.0.0 restructuring with common version control repositories enables version updates of GMAO libraries
such as MAPL to be seamless and improves software collaboration between GCHP and GMAO developers by ensuring that
future improvements in either GMAO or the GCHP community are immediately available to both sets of developers. As an
example, the update to a recent version of MAPL increased optimization of the ExtData layer used for inputs to enable
efficient parallelization to extend from hundreds to thousands of cores, while GCHP updates to MAPL such as bug fixes in
the stretched grid feature have enhanced GMAO capabilities. Updating GCHP to share its infrastructure code with GMAO
via common version control repositories successfully achieves synergistic model development. GCHP's use of forks as Git
submodules allows simple merging between GEOS-Chem and GEOS-ESM GitHub repositories; we utilize GitHub pull
requests and issues for cross-communication and collaboration between GCHP and GMAO developers. Finally, use of
GitHub issues and notifications improves transparency and communication with GCHP users, with all code exchanges and
issues publicly viewable and searchable.





## 7 Improvements to GCHP performance and portability

Here we describe efforts to reduce the difficulty of compiling and running GCHP by improving the build system, the ease of installation through a package manager and software containers, diagnostics, parallelization, and the multi-node cloud capability. Previously, compiling GCHP was more difficult than compiling GEOS-Chem Classic because of the additional
dependencies and complex interface. These surplus codes increased the time required to compile the GCHP model and increased the code brittleness by introducing points of failure. We addressed these issues through a build system generator (CMake) which simplifies building GCHP once its dependencies are satisfied, a package manager (Spack) which automates the process of acquiring missing dependencies, and software containers which can sidestep the entire process in environments that support containers.

### 7.1 CMake

Maintaining a portable and easy-to-use build system is challenging in the context of high performance computing (HPC) software because of the diversity in HPC environments. Environment differences between clusters include different combinations of dependencies in the software stack such as versions and families of compilers, differences in the build-time options of those dependencies such as library support extensions, and differences in system administration such as the paths
to installed software. In practice, these environment differences translate to different compiler options. The Make build system previously used in GCHP was brittle and laborious to maintain due to its need for detailed customization to accommodate differences between clusters. Compared to Make, CMake has a more formal structure for organizing projects and specifying build properties; this facilitates the organization of GCHP's build files and interoperability of GCHP's build files with those of internal dependencies (dependencies which are built on-the-fly during the GCHP build). The
interoperability of CMake-based projects allowed us to leverage existing build files for MAPL, developed and maintained at the GMAO, for building MAPL within the GCHP build.

To address these issues we implemented a build system generator (CMake, cmake.org) in GCHP to (a) improve the robustness of the build, (b) improve the maintainability of the build system, and (c) make building GCHP easier for end-users. Build system generators like CMake are specifically designed to generate a build system (build scripts) for the system,
according to the compute environment. This new build system follows the canonical build procedure for CMake-based builds: a configuration step, a build step, and an install step. During the configuration step, the user executes CMake in a build directory; CMake inspects the environment and generates a set of Make build scripts that build the model. The build step is the familiar compile step where the user runs Make, and the compiler command sequence is executed. The install step is used to port the built executable into the users experiment directory. In addition to a more robust build, benefits of the new
build system include more readable build logs and fewer environment variables.





## 7.2 Spack

Our next step was to ease the installation of GCHP by specifying precisely how to build GCHP for different versions, configurations, platforms, and compilers through an instruction set (i.e., "recipes") for GCHP dependencies which can be built using Spack (Gamblin et al., 2015; http://spack.io/). Spack is an innovative package manager designed to ease
installation of scientific software, by automating the process of building from public repositories all of the dependencies necessary for GCHP if any are missing from the target machine (C compiler, Fortran compiler, NetCDF-C, NetCDF-Fortran, MPI implementation, and ESMF). The flexibility of Spack facilitates implementing numerous build options that can handle a diversity of compilers and environments, thus ensuring that most users can build a functioning copy of GCHP dependencies in a single step, without requiring the user to understand the details of configuring GCHP for their environment. This activity
includes testing the GCHP code with multiple versions of: GNU and Intel compilers; ESMF; multiple MPI implementations (e.g. OpenMPI, MVAPICH2, and MPICH); and the NetCDF libraries. Working configurations are implemented as publicly-available Spack packages to enable new users to install GCHP without concern for conflicts between different versions of different dependencies. The new CMake library is a part of this Spack package.

As part of this effort, we developed a Spack recipe that in a single command allows users to download from the Spack
GitHub repository all GCHP dependencies and build GCHP. Users can modify this command to provide to GCHP specific compile-time build options such as whether to include a specific radiative transfer model (RRTMG; Iacono et al., 2008). Spack also provides syntax for specifying different release versions and compiler specifications for packages and their dependencies. Since GCHP can be built without any proprietary software, open source compilers are sufficient. The GCHP Spack package is maintained by the GEOS-Chem Support Team.
Spack is most useful on systems where few of GCHP's required libraries already exist, e.g. new scientific computing clusters, cloud environments, or container creation. Spack itself only requires a basic C/C++ compiler and a Python installation (since Spack is written in Python) to begin building GCHP's dependencies. These are usually available as standard in most modern Linux environments.

Users can manually specify any existing libraries on their system through Spack configuration files to avoid redundant
installations of GCHP dependencies. This is a required setup step for using existing job schedulers such as Slurm on a user's system. Additionally, Spack's install command includes an option to only install package dependencies without installing the package itself. This option allows users to build and load all dependencies while retaining the ability to modify GCHP source code locally before compiling.

## 7.3 Containers

Both the improvements to the build system and to the installation process are beneficial to most users on most platforms. For HPC clusters that support containers, and for GCHP users in cloud environments, the GEOS-Chem Support Team now maintains pre-built software containers (Kurtzer et al., 2017; Reid and Randles, 2017) containing GCHP and its



dependencies. Software containers provide collections of prebuilt libraries to users that allow GCHP and its software environment to be moved smoothly between cloud platforms and local clusters, so that the identical compute environment

can be executed on any machine. A software container encapsulates a compute environment (the operating system, installed libraries and software, system files, and environment variables) allowing the compute environment to be downloaded and executed virtually on other machines, but without the performance penalty associated with emulating hardware.

We created scripts to automatically generate containers for every new GCHP release, which include GCHP and its dependencies (built using Spack). Users can run one of these containers through Docker or Singularity (which natively

supports running Docker images). Singularity is often preferred for running HPC applications like GCHP because it does not require elevated user privileges.

Software containers are particularly useful for quickly setting up GCHP environments. The only requirements for running one of these containers are the container software (e.g. Singularity) and an existing MPI installation on a user's system. With these requirements met, the only steps needed to run a GCHP container are downloading the container, downloading GEOS-

Chem input data, and creating a run directory.

The main drawback of containers is that many HPC environments do not support their use. Running GCHP with container-based virtualization also results in a 5% to 15% performance decrease compared to an identical build of GCHP run natively on a system. This slowdown results from both additional overhead from using Singularity or Docker and from a lack of system-optimized fabric libraries in the container images.

## 7.4 Error and output diagnostics


Here we describe workflow improvements related to error logging and output. A major challenge for MPI Fortran software is the lack of a standard solution for error logging that exists for other languages. Thus errors in GCHP were difficult to diagnose and debug. To address these challenges, MAPL was extended to include pFlogger (https://github.com/Goddard-Fortran-Ecosystem/pFlogger), an MPI-aware Fortran logging system analogous to Python's "logging" package

(https://docs.python.org/3/howto/logging.html). MAPL users initialize pFlogger with a configuration file (YAML) read at runtime that controls how various diagnostic log messages are activated, annotated, and ultimately routed to files. Per-component log verbosity can then be set to activate fine-grained debugging diagnostics or to suppress everything except serious error conditions. MAPL error messages are all routed through pFlogger and can be optionally annotated to include the MPI process rank and component name and/or split into a separate file for each MPI process.

Output diagnostics that were straightforward in GEOS-Chem Classic are more challenging in GCHP due to distribution of information across processors. A common application of GEOS-Chem has been to compare simulated performance with observations from aircraft campaigns or with monthly means of observations, because differences between the simulation and observation can identify deficiencies in the model or in scientific understanding of the atmosphere. However, GCHP originally could only output data which covered either the entire global domain or a contiguous subdomain. Samples along

aircraft tracks needed to be extracted during post-processing, meaning that users had to store unnecessary data in the interim,

suffering both a performance penalty and a data storage penalty. To facilitate comparisons with observations, we add 1D output capability that allows the user to sample a collection of diagnostics according to a 1-dimensional time series of geographic coordinates, and monthly average diagnostics that account for the variable duration of each month.

## 7.5 Parallelization improvement

The original version of GCHP (Eastham et al., 2018) was well parallelized for simulations on up to several hundred cores (Eastham et al., 2018), and up to 1152 cores on the AWS cloud using Intel-MPI or the elastic fabric adapter (EFA) for internode communication, but suffered from a bottleneck in data input that would significantly degrade performance on a larger number of cores as described in section 3.1. Here we assess the parallelization of GCHP version 13.

We conduct 7-day timing tests on four HPC clusters: Pleiades (NASA), Amazon Web Services (AWS) EC2, Compute1
(Washington University), and Cannon (Harvard University). We focus on typical resolutions at which GCHP is run: C48, C90, C180, and C360. All four clusters use an identical model configuration, except for the number of physical cores per node. The architecture of each cluster is summarized in Table 2. We also compare MPI options and Fortran compilers.

**Table 2: Summary of architectures used to evaluate GCHP performance.**

|  | Pleiades | AWS EC2[1] | Compute1 | Cannon |
|---|---|---|---|---|
| CPU | Intel® Xeon® E5-2680v4 | Intel® Xeon® Platinum 8124M | Intel® Xeon® Gold 6154 | Intel® Xeon® Platinum 8268 |
| Physical cores per socket | 14 | 18 | 18 | 24 |
| Sockets per node | 2 | 2 | 2 | 2 |
| Clock speed | 2.4 GHz | 3.00 GHz | 3.00 GHz | 3.50 GHz |
| Microarchitecture | Broadwell | Skylake | Skylake | Cascade Lake |
| L2/L3 cache size | 256K/35840K | 1024K/25344K | 1024K/25344K | 1024K/36608K |
| Interconnect | InfiniBand | AWS EFA | Infiniband | Infiniband |
| Storage | Lustre | AWS EBS | IBM GFPS | Lustre |

[1]AWS EC2 instances used c5n.18xlarge instances.

Figure 3 shows timing test actual "wall" times. Variability across clusters reflects the effects of different architectures on performance. Tests at C180 resolution exhibit excellent scalability, with near ideal speedup across all systems up to at least a thousand cores. Tests at C90 and C48 similarly exhibit good scalability, albeit with some degradation when using several hundred cores; such large core counts at those coarse resolutions result in excessive internode communication for advection

relative to computation within the node (Long et al., 2015). Tests at C360 resolution conducted on Pleiades demonstrate excellent scalability to 2304 cores, achieving 20 model days per wall day. The performance of different Fortran compilers depends on architecture, with better performance using Intel on Pleiades and better performance using GNU on Compute1. Performance on AWS is better using IntelMPI than OpenMPI at this time.

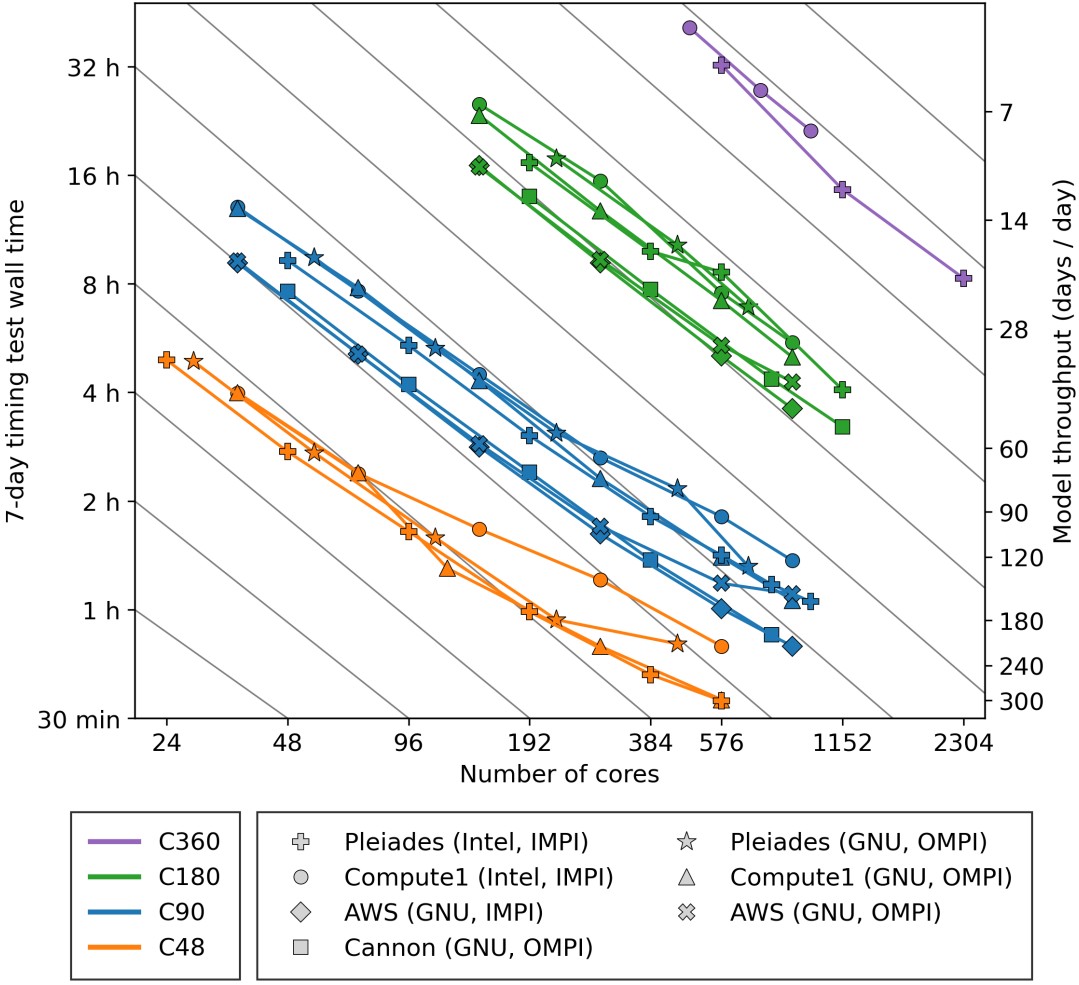

**Figure 3: Timing test results for GCHP version 13 at variable resolutions on multiple platforms. Grey lines indicate ideal scaling. The Fortran compiler and MPI type are indicated in parentheses, with the latter abbreviated as IMPI (IntelMPI) and OMPI (OpenMPI).**





Figure 4 shows a component-wise breakdown of wall times on the Cannon cluster with GNU compilers. Chemistry is the dominant contributor to runtime, as previously shown by Eastham et al. (2018). At C90 resolution with 192 cores, the

GEOS-Chem gridded component (dominated by chemistry) accounts for 84% of the total wall time. This could be addressed in future improvements to the chemistry solver including adaptive reduction of the mechanism (Shen et al., 2021a; Shen et al., 2021b) and smart load balancing to distribute the computationally expensive sunrise/sunset gridboxes across cores and nodes (Zhuang et al., 2020). After chemistry, the next most time consuming component is advection (13%) which also scales well, albeit with some reduction in performance at high core counts that increase inter-processor communication. Data input

now contributes insignificantly to the total wall time. For example, at C90 resolution with 192 cores, data input accounted for only 2.4% of the total wall time. The improvements to the MAPL input server described in section 3.2 resolve the input bottleneck that impaired the original GCHP version.

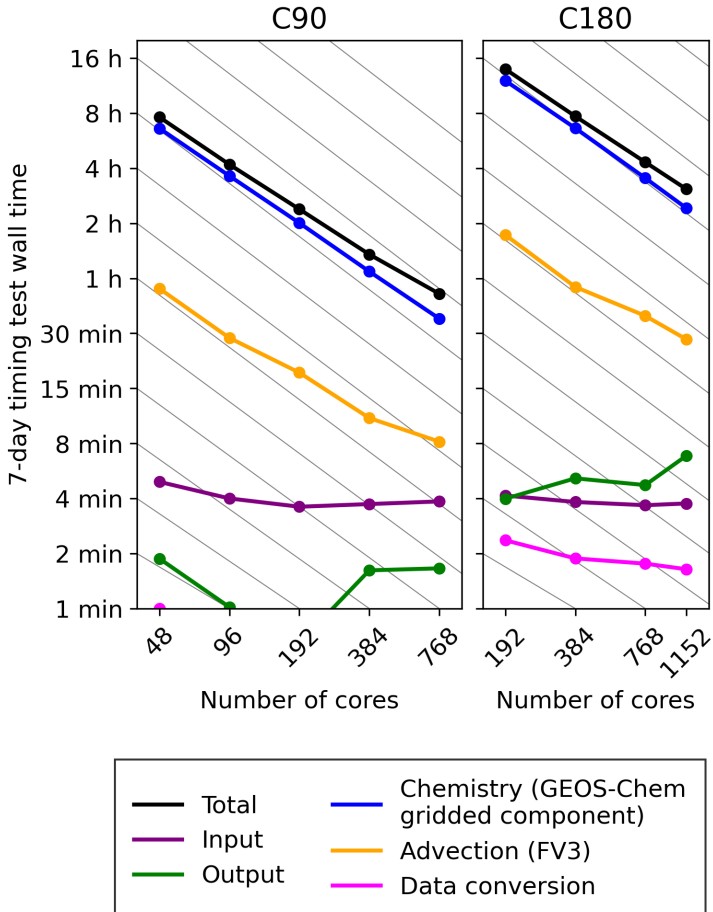

**Figure 4: Component-wise breakdown of the total wall times for timing tests on the Cannon cluster. Grey lines indicate ideal scaling.**



### 7.6. Cloud capability

Cloud computing is desirable for broad community access, for having a common platform where model results can be intercompared, and for dealing with surges in demand that may overwhelm local systems. Zhuang et al. (2020) deployed
GCHP on the AWS cloud for easy user access and demonstrated efficient scalability with performance comparable to the NASA Pleiades supercomputer. In doing so they solved the long-standing problem of inefficient inter-node communication in the cloud (Roloff et al., 2017; Salaria et al., 2017) by using the new EFA technology now available on the AWS cloud. Zhuang et al. (2020) demonstrated the efficient scalability of GCHP on the AWS cloud on up to 1152 cores.

The basic form of a multi-node cluster on the AWS cloud as described by Zhuang et al. (2020), uses a single Elastic Block
Store (EBS) volume as a temporary shared storage for all nodes. The software environment for the main node and all compute nodes is created from an Amazon Machine Image (AMI). The user logs into a main node via SSH and submits jobs to compute nodes via a job scheduler. The compute nodes form an Auto Scaling group that automatically adjusts the number of nodes based on the jobs in the scheduler queue. After finishing the computation, the user archives select data to persistent data storage (S3) and subsequently terminates the entire cluster.
Subsequent to Zhuang et al. (2020), EFA errors at AWS disabled the GCHP cloud capability. We identified specific conditions that cause the failure; we removed settings from the default configuration which were typically extraneous and leading to the failure. We advise users of the specific settings that will result in a failure in the user documentation. Work is in progress to fully restore the capability. Nonetheless, given the stability of the interim solution, GCHP benchmark simulations to assess model fidelity are now routinely being conducted by the GEOS-Chem Support Team on the AWS
cloud.

### 8. Development and application of GEOS cubed-sphere archives

The GEOS operational meteorological archives (GEOS-FP and MERRA-2) have historically been provided only on a rectilinear latitude-longitude grid, rather than on the native cubed-sphere grid. This was intended to facilitate general georeferencing use of the GEOS data but is a drawback for GCHP because of its need to convert the latitude-longitude data
back to the cubed-sphere grid during input at runtime leading to errors through regridding and restaggering. In addition, the previous operational archives included only horizontal winds rather than air mass fluxes, so that advection in GCHP required a pressure fixer to reconcile changes in air convergence and surface pressure (Jöckel et al., 2001; Horowitz et al., 2003). Here we describe the capabilities to directly use (a) mass fluxes instead of winds and (b) data on the cubed-sphere instead of latitude-longitude grid. We then describe two new archives: (a) an operational hourly archive at C720 (~12 km) resolution
and (b) an hourly long-term archive at C180 resolution over 1998-present. We begin with an assessment of mass flux archival on the cubed-sphere. We then describe the data streams being generated, and their archival by the GEOS-Chem Support Team for access by GCHP users.





### 8.1. Mass fluxes versus winds

Standard meteorological archives include time-averaged horizontal winds and changes in surface pressure over the averaging
time period of the archive, typically a few hours. A long-standing source of error in offline models has been the need to use
the archived wind speeds to estimate the air mass fluxes between cells. As the pressure changes over the averaging time
period, the instantaneous wind carries variable mass that is not captured by the wind speed average. In other words, the
convergence computed from the time-averaged winds is not consistent with the archived change in surface pressure.
Perfectly correcting for this error is impossible (Jöckel et al., 2001) although can be compensated for in offline models such
as GEOS-Chem Classic by adjusting the winds with a so-called pressure fixer (Prather et al., 1987; Horowitz et al., 2003).
But it can result in large error in vertical mass transport, which is inferred from the horizontal winds and the change in
surface pressure. The problem can be solved by including air mass fluxes as part of the meteorological archive, but to our
knowledge this had not previously been done for operational meteorological data products.

Figure 5 illustrates the error in surface pressure as computed from air mass convergence using either winds or air mass fluxes
archived from a test GEOS C90 archive over a 5-minute timestep. The top panel shows the surface pressure tendency from
the archive. The middle panel shows the error in this quantity when computed from the archived mass fluxes. The bottom
panel shows the error when computed from the archived winds. We find that using mass fluxes directly rather than inferring
them from wind data reduces the mean absolute error in the surface pressure tendency from 15 Pa to 1.0 Pa. Remaining
errors reflect differences from water evaporation and precipitation that are implicitly included in the pressure tendency
derived from the meteorological data.

The use of air mass fluxes in the meteorological archive requires a new approach for regridding. Mass fluxes are defined
across grid cell edges, rather than at the cell center or averaged over the cell, with basis vectors that change across faces of
the cubed-sphere. Thus if a simulation must be performed at a coarser resolution than the input data, typical regridding
strategies such as area-conserving averaging or bilinear interpolation are not appropriate. Instead, for simulations performed
at a resolution which is an integer divisor of the native-resolution data (e.g. C90 or C180 for C360), fluxes are summed. This
is because the total flux across the edges of a grid cell at coarse resolution is the sum of the fluxes across the coincident
edges of grid cells in the native-resolution data. Fluxes across cell edges which are not coincident are ignored, as these
correspond to "internal" fluxes. We address this integer regridding need through a new capability for MAPL as noted in
Section 3.1.

A related source of error in the original version of GCHP arose from the treatment of moisture in air mass fluxes. The
original version of GCHP computed dry air mass fluxes for advection from winds and "dry pressures", which needed to be
estimated from the surface pressure and specific humidities supplied by GMAO. To reduce this error source, we implement
into GCHP the capability to use total air mass fluxes for advection directly, thus eliminating the need for conversion.



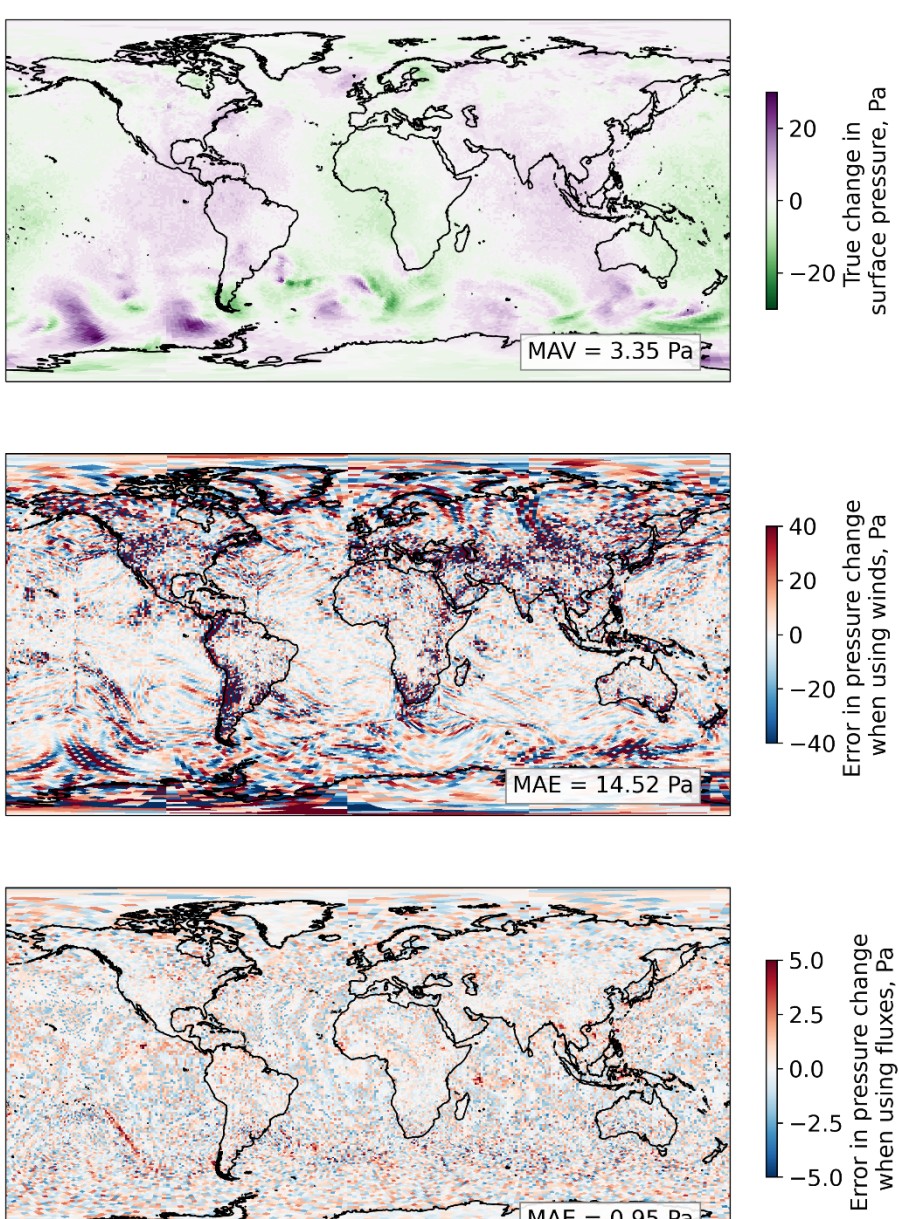

**Figure 5: Illustration of the error in surface pressure change when computed from air mass convergence in an offline model using archived air mass fluxes or winds. Top: true change in surface pressure over a 5-minute time step as computed in a GEOS meteorological simulation at C90 resolution for July 1, 2019. Mean absolute value (MAV) is inset. Middle: error in the pressure change when computed using the archived air mass fluxes from that GEOS simulation. Mean absolute error (MAE) is inset. Bottom: error when the pressure change is computed from the archived winds. Note change in scale.**



## 8.2. Regridding and restaggering

Another source of error is the regridding and restaggering of advection data vector fields to latitude-longitude winds from the cubed-sphere mass fluxes and vice versa, operations that do not preserve the divergence of the vector field. Here we refer to wind as the advection data in an unstaggered grid formation (A-grid) with basis vectors North and East, and mass flux as the advection data in a staggered grid formation (C-grid) with local basis vectors which are perpendicular to the interfaces of the simulation grid-cells. Regridding changes the collocated grids of the vector compnents (i.e., A-grid) from latitude-longitude to cubed-sphere. Restaggering changes the grids of the vector components themselves; in an A-grid the grids of the vector components are collocated and identical to the simulation grid, but in a C-grid the grids of the vector components are distinct and are located at the interfaces of the discrete volumes of the simulation grid. Conceptually, the difference between a vector field on an A-grid and a C-grid is familiar as the distinction between wind (air flow in the North and East directions, defined at one location) and mass flux (air exchange between the finite volumes of the simulation grid, not defined at a single location).

To evaluate the effects of a C-grid cubed-sphere advection data (i.e., mass fluxes) versus A-grid latitude-longitude advection data (i.e., winds), we compare calculations of vertical air mass fluxes, $J_z$. Vertical mass fluxes are expected to be particularly sensitive to errors because they are computed from the convergence of horizontal mass fluxes. We use advection input data archives on a C180 cubed-sphere with C-grid mass fluxes and a $0.5°×0.625°$ latitude-longitude grid with A-grid winds. Both archives were generated by the same GEOS simulation, which had a native grid of C180. All quantities on the A-grid are defined at the center of the grid cell, including both components of the wind vector, while on the C-grid the two components of the air mass flux vector are evaluated at the center of the relevant cell edge. We compare three alternative calculations of vertical mass fluxes:

$J_z(MF_{CS})$ is the vertical mass flux computed using native C180 C-grid mass fluxes; the C-grid mass fluxes are neither regridded nor restaggered.

$J_z(Wind_{CS})$ is the vertical mass flux computed using C180 A-grid winds; the original C-grid air mass fluxes are converted to winds and restaggered from C- to A-grid, and then restaggered from A- to C-grid when they are loaded in GCHP. The operations involve restaggering but no regridding.

$J_z(Wind_{LL})$ is the vertical mass flux computed using $0.5°×0.625°$ A-grid winds; the original C-grid air mass fluxes are converted to winds on the latitude-longitude A-grid, and regridded from $0.5°×0.625°$ to C180 and then restaggered from A- to C-grid when they are loaded in GCHP. The operations involve both restaggering and regridding.

The operations performed to the input data for $J_z(MF_{CS})$, $J_z(Wind_{CS})$, and $J_z(Wind_{LL})$ are summarized in Table 3.

Figure 6 compares the vertical mass flux calculations in the lower troposphere (near 900 hPa), mid-troposphere (near 500 hPa), and mid-stratosphere (near 50 hPa) for a 5-minute timestep at a nominal time (2017-03-01 12:30). In the troposphere, $J_z(Wind_{LL})$ and $J_z(Wind_{CS})$ both exhibit dampened upward and downward motion compared to $Jz(MF_{CS})$, as well as spurious noise. The dampening and noise in $J_z(Wind_{CS})$ is significantly less than in $J_z(Wind_{LL})$, which is consistent with the extra



regridding operations done to the $J_z(Wind_{LL})$ input data. The comparison of $J_z(Wind_{CS})$ and $J_z(MF_{CS})$ in the right column of

Figure 6 demonstrates that restaggering, even on the native grid, weakens vertical advection (slope=0.92). In the stratosphere where vertical motion is weak, both $J_z(Wind_{LL})$ and $J_z(Wind_{CS})$ are dominated by noise, reinforcing the importance of mass fluxes for vertical transport processes in the stratosphere.

**Table 3: Operations applied to GEOS advection data for input to GCHP[a].**

|  | MF$_{CS}$ | Wind$_{CS}$ | Wind$_{LL}$ |
|---|---|---|---|
| GEOS data egress operations | None | C[b] →A[c] restaggering (S)[d]<br>Change of basis (M)[e] | C →A restaggering (S)<br>Change of basis (M)<br>CS → LL regrid (S) |
| GCHP data ingress operations | None | Change of basis (M)<br>A → C restaggering (S) | LL → CS regrid (S)<br>Change of basis (M)<br>A→ C restaggering (S) |

[a]The GEOS native data are cubed-sphere mass fluxes on the C-grid (MF$_{CS}$) but are then converted in the standard archive to

latitude-longitude winds on the A-grid (Wind$_{LL}$). The GCHP model reconverted these Wind$_{LL}$ data to MF$_{CS}$ for input.

[b]Quantities on the C-grid are defined at the center of the relevant cell edge.

[c]Quantities on the A-grid are defined at the center of the grid cell.

[d]Operations that are a systematic source of error are marked with (S) and operations with machine-precision are marked with (M).

[e]Basis vectors differ for winds versus mass fluxes.

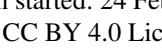

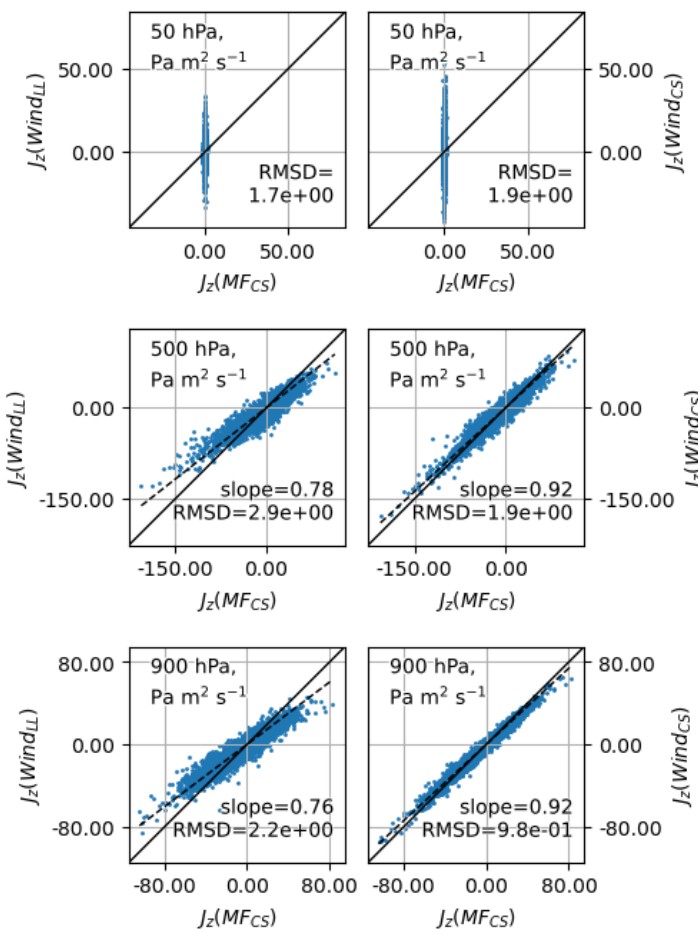

**Figure 6: Comparison of vertical mass flux calculations at 50 hPa, 500 hPa, and 900 hPa in the global GCHP domain using different input fields for a 5-minute timestep at an example time (2017-March 1 12:30:00). Each point represents a grid cell at the corresponding pressure. *Jz(MF$_{CS}$)* is the vertical mass flux using native C180 C-grid mass fluxes. *Jz(Wind$_{CS}$)* is the vertical mass flux using C180 A-grid winds. *Jz(Wind$_{LL}$)* is the vertical mass flux calculated using 0.5° × 0.625° A-grid winds. Note different scales for the different rows of panels.**

### 8.3 Archive descriptions

Given the importance of cubed-sphere air mass fluxes for accuracy in off-line advection computations, and the previously noted need for higher temporal resolution to avoid smoothing of eddy and convective motions (Yu et al., 2018), two new cubed-sphere archives with hourly resolution are now being generated at GMAO as part of the GEOS-FP and GEOS-IT data streams. The generation of a new cubed-sphere GEOS-FP meteorological archive as a manageable operational product at GMAO is however a challenging task due to the additional output costs on top of the computationally intensive GEOS system. Despite the C720 resolution of the GEOS-FP system, the current operational archive is produced at 0.25°x0.3125° resolution (corresponding to C360) with 3-hourly 3D fields including winds, because of output limitations.





We overcome this operational hurdle in GEOS-FP by limiting the hourly production of C720 output to the advection variables, and having those archived by the GEOS-Chem Support Team on the Washington University cluster. The cubed-sphere archive is most critical for advection variables. Other meteorological variables can be conservatively regridded from the operational 0.25°×0.3125° archive. Advection requires only two 3D variables in the hydrostatic atmosphere of the GEOS

system, namely the horizontal air mass fluxes and Courant numbers (to determine the number of substeps in the FV3 advection calculation), and 2D surface pressure. Currently the specific humidity is also archived, to allow accurate conversion between dry and total mass mixing ratios. This operational production of hourly C720 advection output has been ongoing in GEOS-FP since March 11, 2021, and this output is continuously being archived by the GEOS-Chem Support Team.

GMAO is also generating an hourly C180 full cubed-sphere GEOS-IT archive for all variables for the period 1998-present. This GEOS-IT archive will offer long-term meteorological consistency akin to the MERRA-2 archive, but on the cubed-sphere using GEOS-5.29. Both mass fluxes and winds are being archived. Two-dimensional products are also being provided on a latitude-longitude grid. This offline GEOS simulation offers the capability to archive the entire cubed-sphere dataset without the constraints of an operational system. Completion of the entire 24+ year archive expected by early 2023.

8.4. Direct ingestion of GMAO meteorological data

GEOS-Chem has historically required reprocessing the GEOS meteorological data from GMAO into suitable GEOS-Chem input files. This reprocessing included modifying certain fields such as cloud optical depth into formats expected by GEOS-Chem, regridding data to coarser resolution as required by GEOS-Chem Classic, extracting regional data for pre-defined nested simulations, and flipping the vertical dimension of the arrays. We have developed the capability for GCHP to directly

use the GMAO meteorological archive without modification and this will be an option in the standard model (version 13.4.0). This capability not only reduces effort, data duplication, and possible errors, but also facilitates simulations at near real time that directly read the operational post-processing and forecast data produced by GMAO.

## 9 Demonstration of technical performance

We bring together the developments described above to demonstrate the technical performance offered by GCHP. Figure 7

shows a GCHP simulation of tropospheric $NO_2$ columns for April 9-15, 2021 using mass fluxes from the new hourly C720 GEOS-FP operational archive. Pronounced heterogeneity is apparent in tropospheric $NO_2$ column concentrations, with clear enhancements over major urban and industrial regions. The attributes of high resolution are apparent for example along western South America, where the C720 resolution resolves distinct urban areas of Chile, Argentina, and Peru that were not evident at coarser resolution. Global population-weighted $NO_2$ column concentrations simulated at C720 are twice those at

2°x2.5°, indicating the importance of the high resolution offered by GCHP for air quality assessments. Table 4 contains statistics describing the simulations shown in Fig. 7, as conducted on the Pleiades cluster. The GCHP full chemistry simulation on 224 million grid boxes using 2,904 cores achieved a throughput of 5.2 days/day.



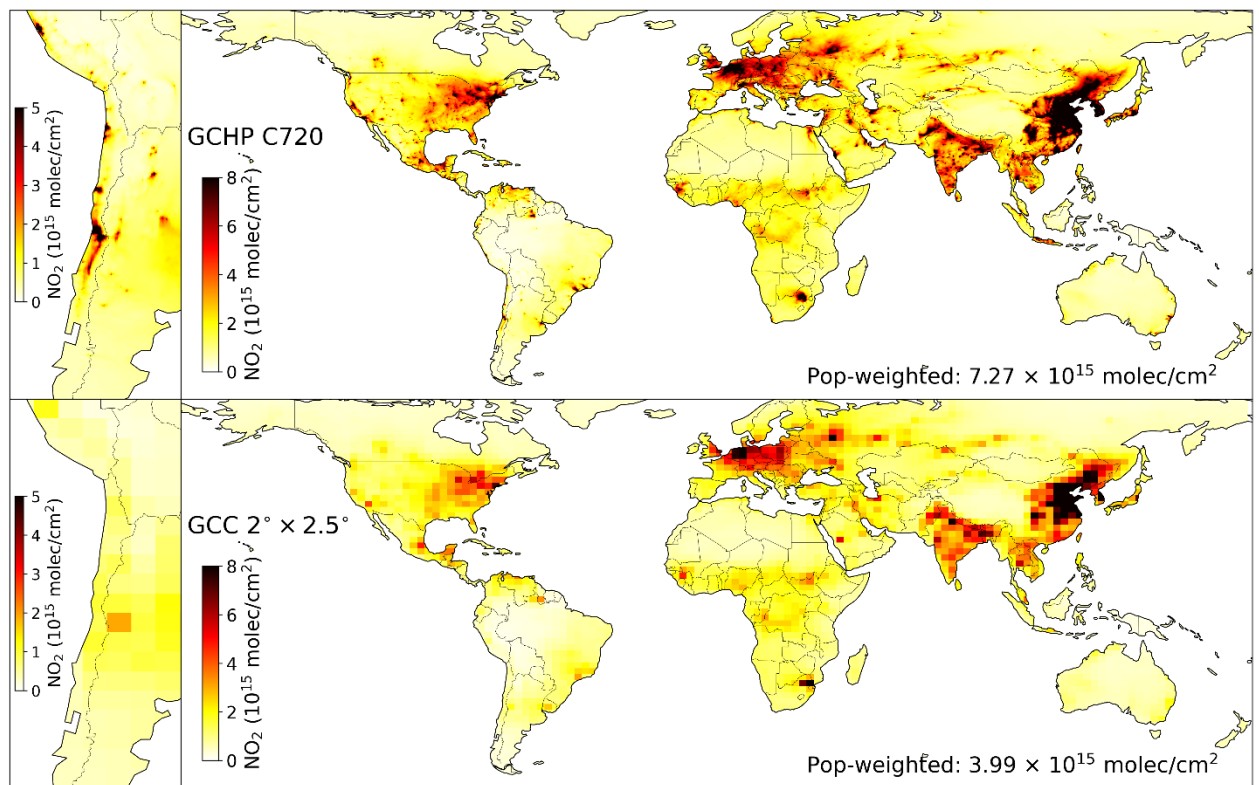

**Figure 7: Simulations of tropospheric NO₂ columns using GCHP at C720 resolution (top) and GEOS-Chem Classic (GCC) at 2º x 2.5º resolution (bottom). Both simulations used version 13.2.1 for the period April 9-15, 2021 following a one-week spinup.**

**Table 4: Characteristics of GCHP and GEOS-Chem Classic (GCC) simulations[a] in Fig. 7**

| GEOS-Chem simulation | Total number of grid cells | Operator durations (minutes) | Number of physical cores | Total wall time (days) | Throughput (days/day) |
|---|---|---|---|---|---|
| GCC $2° \times 2.5°$ | 943,488 | C20T10 | 36 | 0.27 | 52.4 |
| GCHP C720 | 223,948,800 | C10T5 | 2,904 | 2.69 | 5.20 |

[a]Simulations for April 2-15 were conducted on the Pleiades cluster. See Table 2 for the cluster architecture.

[b]Operator durations are represented as C$c$T$t$ where $c$ is the chemical operator duration and $t$ is the transport operator duration.

## 10 Future needs and opportunities

The developments described above and now made available through the GEOS-Chem version 13 series increase the accessibility, accuracy, and capabilities of GCHP, but also highlight future opportunities for improvement. We identify here





four key opportunities to 1) further improve GCHP accessibility including on the cloud, 2) develop a tool for GCHP integration of satellite observations, 3) increase GCHP computational performance, and 4) modularize GCHP components.

**Improve GCHP accessibility including on the cloud.** There are four main areas where GCHP accessibility could be improved to benefit users. (a) Current GCHP configuration files are complicated with 12 input files, 10 file formats, redundant specification, and platform-specific settings. The need remains to simplify the process of configuring a GCHP

simulation by consolidating the number of user-facing configuration files, eliminating overlap, and reducing the number of file formats. (b) The meteorological and emission input data for GCHP are extensive with over one million files available. It is challenging for users to identify and retrieve a minimus set of files needed for their simulation. This issue could be addressed with a cataloging system such as STAC (stacspec.org) which is specifically designed for cataloging Earth system data. (c) Analyzing GCHP output is currently impeded by its large data volumes. The next generation of file formats for

Earth systems data such as Zarr (zarr.readthedocs.io) offers opportunities to efficiently index GCHP output data during analysis. (d) The process of setting up GCHP on the cloud is labor intensive. This could be addressed with automated pipelines for environment creation, input data synchronization, execution, and continuous testing. These developments would facilitate user exploitation of the full resources of GCHP for simulations of atmospheric composition.

**Develop tool for GCHP integration of satellite observations.** Quantitative analyses of satellite observations with a CTM

require observational operators that mimic the orbit tracks, sampling schedule, and retrieval characteristics of individual satellite instruments. Developing these observational operators is presently done in an ad hoc way in the GEOS-Chem community, resulting in duplications of effort and representing an obstacle for the exploitation of satellite data. A general facility to which the community could readily contribute would allows users to select satellite orbit tracks and instrument scan characteristics, and to apply instrument vertical sensitivity such as through air mass factors and averaging kernels. This

open-source library of observational operators would increase the utility of GCHP for interpretation and assimilation of satellite observations.

**Increase GCHP computational performance.** Chemistry is the most time-consuming component of GCHP calculations and remains a major barrier to the inclusion of atmospheric composition in ESMs. Two general bottlenecks currently impede performance, in GCHP and other atmospheric composition models: a) unnecessarily detailed chemical calculations in

regions of simpler chemistry such as the background troposphere or stratosphere, and b) idled processors awaiting completion by a few processors of lengthy calculations at sunrise/sunset. The first bottleneck could be addressed by applying recent developments in adaptive chemical solvers for greater efficiency, and the second through smart load balancing that more efficiently allocates processors across grid boxes, thus enabling high-resolution global simulations with complex chemistry.

**Modularize GCHP components.** GCHP consists of a number of operators computing emissions, transport, radiation, chemistry, and deposition. Modularization of these operators will facilitate exchange of code with other models for both scientific benefit and good software engineering practice. This has already been done with the emissions component (HEMCO), which is now adopted in the NASA GOCART, NCAR CESM, and NOAA GFS models (Lin et al., 2021). There



is a strong need to generalize this practice to other GCHP modules. This will avoid redundancy and promote interoperability
with other atmospheric composition models used by the research community, including in particular in the NASA GEOS
system.

*Code Availability*. GCHP is publicly available at www.geos-chem.org with documentation at gchp.readthedocs.io. The latest
GCHP version (13.3.4) is available at zenodo.org/record/5764877#.Ydr54_7MI2w.


*Data Availability*. GEOS-FP output is publicly available at https://fluid.nccs.nasa.gov/weather/ and from an archive
maintained by the GEOS-Chem Support Team at http://geoschemdata.wustl.edu.

*Author contributions*. RVM, DJJ, and SDE conceptualized the project. RVM, DJJ, SDE, TLC, CAK, and SP conducted
funding acquisition. SDE, LB, EWL, TLC, CAK, WD, DZ, RAL, MPS, RMY, YL, LE, WMP, BMA, and ALT contributed
to development of GCHP version 13 code and input data. LB, SDE, EWL, and DZ conducted data analysis and visualization.
RVM, DJJ, SDE, LB, EWL, TLC, CAK , and WD wrote the manuscript with contributions from all coauthors.

*Competing interests*. The authors declare that they have no conflict of interest.


*Acknowledgements*. This work was supported by NASA AIST Grant 80NSSC20K0281.  Resources supporting this work
were provided by the NASA High-End Computing (HEC) Program through the NASA Advanced Supercomputing (NAS)
Division at Ames Research Center, and by local computing platforms at Washington University and Harvard University.

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
