# Peer review of "Improved Advection, Resolution, Performance, and Community Access in the New Generation (Version 13) of the High Performance GEOS-Chem Global Atmospheric Chemistry Model (GCHP)"

_Geoscientific Model Development, 2022_

## Author Response (AR1)

**Review Responses for:** Improved Advection, Resolution, Performance, and Community Access in the New Generation (Version 13) of the High Performance GEOS-Chem Global Atmospheric Chemistry Model (GCHP)" by R. V. Martin et al.

We thank both referees for their comments, which have helped improve the quality and clarity of our manuscript. We have responded to each comment below. The original comments are in black, our responses are in blue and the changes to the manuscript text are in *blue italics*. Overall, the manuscript conclusions remain unchanged, but we have reordered the manuscript and reduced repetition as suggested.

**Referee 1**

**General Comments:**

The paper outlines the developments under GEOS-Chem version 13 series to increase the accessibility, accuracy, and capabilities of the global atmspheric chemistry model focusing on the new generation high-performance GEOS-Chem (GCHP). Overall, a number of significant improvements are described and quantified spanning the model performance, ease of use, and resolution/accuracy.

The Introduction section provides an overview fundamental concepts with respect to atmospheric composition modelling, with particular focus on Chemical Transport Models and the development of GEOS-Chem.

In general there is a lot of repeition of listing the novelties at the end of the Introduction, again in Section 4 (in particular in Table 1) and in the beginning of different sections (e.g. Sec.7). The paper can be streamlined by making a single listing and then directly going into the details in sub-sections.

We have streamlined the manuscript by treating Section 4 as the single listing. We have removed repetitive text at the end of the Introduction and at the beginning of Section 7 (now section 8).

Section 6 can perhaps benefit in clarity by being shortened and made more succinct, or even removed altogether/moved to a supplement as it contains software engineering details that are esoteric and may not be of relevance to a general audience, and elaborates on problems of previous releases.

We have shortened and moved section 6 later (to section 7) so that the software engineering content is less distracting. We respectfully retain this section since it is fundamental to the manuscript.

The the end of Sec. 7, some details on what was removed, or a specificreference to the relevant documentation section/site should be added.

Paragraph rephrased for more specificity and reference added to the documentation.

Finally, it is proposed to restructure the manuscript to have the very important Sections on the numerical calculations and model quality developments and performance assessment first (Secs.8,9), and then follow with the software engineering (containers, build system) aspects.

We have moved section 8 to precede the software engineering developments. We respectfully retain section 9 in its current location since it is a technical demonstration that relies upon all prior sections.

**Specific Comments:**

Comments are prepended by page and line number of the pre-print.

p.4 l.101-104: Please consider rephrasing the sentences to make the meaning more clear (in particular the transport processes).

Sentence rephrased.

p.6 l.6: Not clear what an "unsatisfied" import is. Please explain briefly in text.

Clarified.

"*MAPL automatically aggregates all component exports to make them available to the History component for output. If a parent cannot provide a value for any given import of its children, that import is labelled as "unsatisfied" and is automatically incorporated into the import state of the parent. Any imports that remain unsatisfied at the top of the hierarchy are routed to the ExtData component which attempts to provide values from file data.*"

p.6 l.159-161: The paragraph is not providing specific information. Authors could elabborate on specific gaps in ESMF. Does MAPL presently only provide an additional regridding method?

Added specificity.

"*Currently MAPL provides a regridding method not yet available in ESMF, namely the ability to regrid horizontal fluxes in an exact manner for integral grid resolution ratios. MAPL also extends ESMF regridding options to implement methods that provide for "voting" (majority of tiles on exchange grid wins), "fraction" (what fraction of tiles on exchange grid have a specific value), and vector regridding of tangent vectors on a sphere.*"

Sec. 3.2: Perhaps a table with a synopsis of the resolution, grid-type, time span/step, for the different analyses would assist the reader.

Table added.

p.8 l.211: Instead of just stating "20 model days on 2304 cores", the relative improvement on equivalent hardware would better showcase performance gains. Or even better refer to to the relevant Section 7.5 (Fig.3).

Sentence cut to reduce repetition and focus on referring to the relevant section.

p.9. l.257: "The original version of GCHP was implemented as a code repository" - meaning not clear, please rephrase: Do you mean implemented as separate code base?

Rephrased.

"*The original version of GCHP (Eastham et al., 2018) was implemented as a single code base that was separate from the GEOS-Chem code base, that included copies of supporting libraries such as MAPL and ESMF, and that users needed to manually insert into the GEOS-Chem code base.*"

Fig.4: some results are cut/not visible in the left panel. Though it's clear it's under a minute, the presentation can be improved.

Figure revised to include missing results.

Fig.5 (and Sec.8.1) Is the error in units of Pa or hPa?

Pa as stated.

p.18 l.485: Can the authors quantify the approximate magnitude of the error/improvement regarding mositure in air mass flux?

Unfortunately we are not able to provide an estimate because we only have air mass fluxes which include water.

**Editorial comments:**

Web addresses should be added as references instead of inline.

Done.

p.6 l.168: surface quantities -> surface variables

Done.

p.8 l.208: opportunities -> capabilities

Done.

p.8 l.212: actual time -> wall-clock time

Sentence removed for brevity.

p.8 l.226: use of winds -> use of wind fields

Done.

p.8 l.227: to and from a latitude-longitude grid to and from cubed-sphere grid -> between latitude-longitude and cubed-sphere grids; and similar for restaggering

Done.

p.8 l.234: FlexGrid is only for latitude-longitude -> FlexGrid only supports latitudelongitude

Done.

p.9 l.237: Remove nimble

Done.

Sec. 6 title: Software collaboration -> emgineering

Done.

p.20 l.501: vector compnents typo

Corrected.

p.20 l.505: familiar -> straightforward/simple/elementary

Removed.

**Referee Mathew Evans**

This paper describes recent updates to the "High performance GEOS-Chem" atmospheric chemistry transport model. Some of these updates give updated scientific capacity to the model (stretched grid, improved transport) others are more technical and revolve around the software side of running the model (Cmake/spack, containers, multi-node performance). The paper is a useful reference to both the GEOS-CHem community and more widely the geophysical modelling community.

I think this paper should be published. I make some comments below about potential changes which could improve the paper. I do not need to see the paper before publication.

Thank you.

It might make sense to separate the software engineering sides of the modelling (Cmake, containers, multinode performance) from the "science changes" (stretched grid, improved transport). I think this only involves a reordering of sections and some explanation for the order.

Sections reordered as suggested, and brief rationale added at the end of the introduction.

Table 1 seems to do a similar job to text around line 80. These feel a bit redundant.

Text reduced near line 80.

Figure 3 shows a number of common features of the systems used. Cannon is always slower than Pleiades for a certain number of nodes. It is hard to tell on the log scale of the graph but this appears to be a factor of 2. This seems surprising given the cores per node and clock speed advantages of Cannon and the similarity of interconnect, and file system. It might be useful to provide some commentary on this if there is some understanding of why this is, even if it is speculative.

Text clarified that Cannon is actually faster and explanation added that clock speed and cache are likely drivers.

Figure 6. The units of this seem a bit strange to me. Is this the mass moved across a vertical grid box in a second ie kg s-1? Pa m^2 s-1 appears to be a slightly complicated set of units for this (Pa m^2 s-1= kg m-2 m^2 s-1= kg s-1).

Thanks for catching this.  Units corrected.

Figure 7 is very attractive. It also highlights the need for us to be able to simulate the composition of the atmosphere at high resolution. The population-weighted NO2 is half at high resolution than at low. It would probably be worth putting in a sentence or two to emphasise this.

Added a sentence to the abstract, and additional main text, to increase emphasis.

Line 630. Would the authors like to provide a list of processes which would most benefit from increased modularization?

List added.

Line 21. Transformative is quite a strong word with probably quite a high bar for use. I might think about dropping that word.

Changed transformative to major.

---

## Author Response (AR2)

**Review Responses for:** Improved Advection, Resolution, Performance, and Community Access in the New Generation (Version 13) of the High Performance GEOS-Chem Global Atmospheric Chemistry Model (GCHP)" by R. V. Martin et al.

We thank editor for the comments, which have helped improve the quality and clarity of our manuscript. We have responded to each comment below. The original comments are in black, our responses are in blue and the changes to the manuscript text are in *blue italics*.

**Comments to the author**:

One of the reviewers has now checked your revised submission, and now both recommend publishing your manuscript. After reading it, I agree. However, before accepting it, I think a few issues must be fixed. I list them next:

- First, Spack is a core part of your deployment, and you have adapted a version to be used with GCHP; therefore, I think you must publish it in Zenodo, as you did with the model code. In this way, it is important that you name it with a version number.

We added to the Code Availability section, the link to the Spack post on Zenodo. We also updated the GCHP version link.

- I find it highly interesting that you have tested the model in AWS, and you spend a whole subsection of the manuscript describing it and discussing this capability. However, I think readers would benefit from putting your results into context. A behaviour that has been observed before is that cloud environments can outperform local supercomputers for a low number of cores (see, for example, Montes et al., 2019; I understand that this could be self-serving from my side, but I am not aware of many of these evaluations for climate models). This is probably related to the selection of the cloud infrastructure, where it is possible to allocate a few cores connected with a fibre channel, and when the number of cores exceeds the capability of servers in the same rack, cores allocated could be in a different data centre. Something in this way can be observed in your figure 5 for AWS C90 and C180 (independently of the use of IMPI or OMPI). I think it is important that this is noted in the paper, as it is something that eventually could have a more significant impact on performance than, for example, others that you mention, such as the use of containers.

We added the following at the introduction to this section.

*Cloud computing has been able to outperform local supercomputers for a low number of cores (e.g., Montes et al., 2020), but HPC applications with intensive internode communication have previously not scaled well to a large cluster on the cloud (Mehrotra et al., 2016;Coghlan and Katherine, 2011).*

Please, when you mention Git, cite the work explaining it: Torvalds (2014)

Reference added at line 395 where Git is first mentioned.

---

## Author Response (AR3)

**Review Responses for:** Improved Advection, Resolution, Performance, and Community Access in the New Generation (Version 13) of the High Performance GEOS-Chem Global Atmospheric Chemistry Model (GCHP)" by R. V. Martin et al.

We thank the editor for the comment, which has helped improve the manuscript. We have responded to the comment below. The original comment is in black, our response is in blue.

**Comments to the author**:

I have checked the repositories that you have set up. One says to contain the code for the model; however, it does not contain the model. It only includes a set of scripts and instructions to clone a GitHub repository. This is the same that storing the code stored in GitHub, and we can not accept it.

Therefore, please, include with your manuscript a repository with the complete version of the model used.

We have updated the Zenodo post with a complete repository of the code.